# Effect of Strength vs. Plyometric Training upon Change of Direction Performance in Young Female Handball Players

**DOI:** 10.3390/ijerph19116946

**Published:** 2022-06-06

**Authors:** Hallvard Nygaard Falch, Markus Estifanos Haugen, Eirik Lindset Kristiansen, Roland van den Tillaar

**Affiliations:** Department of Sports Sciences and Physical Education, Nord University, 7600 Levanger, Norway; falch@hotmail.com (H.N.F.); haugen@helseogprestasjon.no (M.E.H.); ek1105@hotmail.com (E.L.K.)

**Keywords:** force, velocity, specificity, COD

## Abstract

The aim of the current study was to investigate the effect of six weeks of strength vs. plyometric training upon change of direction (COD) performance. A total of 21 young female handball players were randomly assigned to either a strength group: (n = 11, age: 17.5 ± 2.3 years, height: 1.69 ± 0.05 m, weight: 65.8 ± 5.9 kg) training bilateral, unilateral and later squats; or a plyometric training group (n = 10, age: 17.1 ± 2.4 years, height: 1.73 ± 0.07 m, weight: 67.1 ± 9.3 kg) training drop jumps, unilateral countermovement jumps and skate-jumps. Groups were assigned after being pair-matched based upon baseline COD performance. The training modalities were matched in training impulse. A force- (180°) and velocity-oriented (45°) COD of 20 m was used to measure changes in COD performance (10 m + COD + 10 m). Total time (s) to complete the COD test was defined as the performance variable. The level of significance was set at *p* < 0.05. The two-way ANOVA showed no group effect upon COD performance. A significant effect was only observed for the strength training group in the last 10 m and total 20 m of the force-oriented COD (F ≥ 5.51; *p* ≤ 0.04; η^2^ ≥ 0.36). Both groups improved performance in other strength- and power-related tests. It was concluded that only the strength training program was effective in developing force-oriented COD performance in the studied population, while the plyometric training program was not sufficient. Both training modalities are useful for improving performance in different strength and power tests in young female handball players.

## 1. Introduction

Team sports consist of multidirectional movements, whereby athletes are required to rapidly accelerate, decelerate, and perform several changes of direction (COD) during a match [1,2,3,4]. The COD is a pre-planned movement of multiple steps, dependent on the athletes physical abilities [5,6], without the cognitive aspect of unplanned ‘agility-manoeuvres´ [7]. The COD consists of an acceleration followed by a deceleration, before re-accelerating into a new direction [7] and is important for the match outcome [8]. The COD ability is task-dependent [5,9], whereby the technical execution, estimated energy expenditure, and muscle activity varies between the different COD tasks, which is distinctive between sports and positions on the field, as well as anthropometrical factors such as limb lengths [6,9,10,11,12]. Independent of sport-distinctive differences, athletes need to express a great amount of force in a short amount of time to perform a rapid COD [8,13]. The required ground reaction forces for braking and the direction of forces in the plant step are determined by the athletes’ initial velocity approaching the COD and the degrees of the pivoting movement [9,14,15].

Higher velocities require greater force for braking produced by eccentric muscle work. Also, CODs of greater angles require a greater magnitude of force to change momentum, applying force for a longer time to create a propulsive force to re-accelerate into a new direction [6,16,17]. Based on the angle of the turn, CODs have been hypothesized to be oriented towards being more force-oriented (>90°), requiring substantial braking forces distributed over several foot contacts to manage the turn. On the other hand, CODs where velocity-maintenance is more prominent (<90°) have been suggested to be velocity-oriented, as there is less change in momentum and forces are more evenly distributed across foot contacts [9,14]. CODs of greater angles has been found to not only affect the magnitude of forces produced in the penultimate step and plant step, but also the direction of forces which is directed more horizontally and expressed at longer foot contacts [16,18]. Due to the greater expression of force and longer foot contact times, force-oriented CODs require a greater impulse to shift the athlete’s momentum, both when decelerating and when re-accelerating.

The hypothesis is based on the Newtonian laws of motion, as athletes must express a net force to the ground to overcome inertia and change direction [6]. As force is a product of mass x acceleration, increasing strength relative to bodyweight is suggested to positively influence COD performance, which might be of increased importance in force-oriented CODs to tolerate the higher loads [18,19,20]. However, traditional training for maximal strength is often performed at slower muscular contraction velocities in comparison to the gait cycles performed in the COD manoeuvre. Although the intention of performing exercises at a fast concentric velocity is also important in exercises with high external loads, the isolated transfer to sport-specific movements is inconsistent [21]. Investigations of transfer to sport-specific tasks such as the COD might be especially important when considering that the velocity-oriented CODs emphasizes velocity maintenance, whereby ground contact times is short [6,22]. The plant and penultimate step of the COD is usually < 0.44 s in velocity-oriented CODs [16] dependent on the COD task and the population of the subjects measured, while force-oriented CODs require longer ground contact times in comparison. Nevertheless, muscular strength is an integral part for producing force, thus an important component for athletes to express high levels of power in dynamic movements (P = F ∗ v) [13]. Furthermore, horizontal accelerations and decelerations have been found to differentiate between fast vs. slow COD performers in young female team sport athletes [23]. Although studies have observed that strength training positively influences COD performance, exercises performed at slow muscular contraction velocities have been suggested to be more specific towards developing force-oriented CODs if not accompanied by high-velocity exercises [9]. The current knowledge of the effect of strength and plyometric training upon enhancing COD performance is limited and greatly stemmed from training interventions conducted in male soccer players [9].

Exercises performed at higher muscular contraction velocities, such as plyometrics, have been found to improve COD performance across a spectrum of force-velocity COD requirements [24,25,26,27,28]. However, only one study by Rædergård, et al. [29] has specifically examined how training at slow (strength) vs. high (plyometrics) contraction velocities affect performance changes in force- vs. velocity-oriented CODs; it suggested that plyometrics are more specific for enhancing overall COD performance. The study was conducted in men adapted to strength training. Contradictorily, Falch, et al. [23] reported that young female athletes’ performance in strength exercises is more associated with both force- and velocity-oriented CODs compared to plyometric exercises. This may be due to a certain amount of strength being needed to express high levels of power output [13] and that there might exist a threshold of strength development before athletes should emphasize plyometric exercises. This contradiction might therefore be a result of the female athletes in the study by Falch, Kristiansen, Haugen, and van den Tillaar [23] expressing lower levels of relative strength, indicating strength training to be a more suitable training in modality in such a population oppose to plyometric training.

With limited time available in which to prioritize COD-specific physical training in team sports, it is important to choose task-specific exercises for the distinctive sport. Thus, the current study aims to investigate how strength vs. plyometric training affects young female handball players performance in both force- and velocity-oriented CODs. As both training modalities seek to improve physical qualities related to power productions, both strength and plyometric training were hypothesized to positively enhance both force- and velocity-oriented COD performance.

## 2. Materials and Methods

A within-subject design with pre-to-post measurements was used to investigate the effect of strength and plyometric training upon force- and velocity-oriented COD performance in female team sport athletes. A between-subject design was used to compare the effect of the two training modalities. To limit the effect of learning, the subjects participated in a familiarization day, during which they practiced all the different tests. Performance changes in total time (s) to complete the force- and velocity-oriented COD test was the dependent variable of the study, while training modality was the independent variable.

### 2.1. Subjects

The team sport athletes in the current study comprised 27 young female handball players recruited from the academy of an elite team playing at the second-highest level in the Norwegian league system. The athletes had to declare the absence of any injury or illness for the last three months which could negatively impact participation. Risks and benefits of participation were explained for all the subjects accompanied by written consent, which had to be signed by the athletes (or their guardians if the athlete was under 18 years old). The study complied with the latest revision of the Declaration of Helsinki and was approved by the Norwegian Centre for Research (project approval: 903955). The subjects were informed to be mentally and physically prepared for the training and testing sessions. A total of six athletes were excluded from the study due to either injury in competition (n = 3) or missing more than two plyometric or strength training sessions (n = 3). Thus, 21 athletes were assigned to either a strength training group (n = 11) or a plyometric training group (n = 10) (Table 1).

### 2.2. Procedures

Time (s) was the performance variable of COD performance (45° and 180°) in the current study. Furthermore, a 30 m sprint and a horizontal braking test were utilized to measure acceleration, and horizontal braking force and power. Three tests were utilized to measure strength by external load lifted (bilateral, lateral, and unilateral squat), as well as three plyometric tests: drop-jump (reactive strength index), skate-jump length, and unilateral countermovement jump height. The different tests were retrieved from earlier research in which such tests were thoroughly described [22,23].

The days of familiarization and the pre- and post-tests were executed in an identical manner, starting with a standardized warm-up protocol by van den Tillaar, et al. [30], before the athletes were randomized into three groups on familiarization day. Subsequently, on familiarization day, the three groups were randomly assigned to start to test performance in one of the type of tests (strength, plyometrics, or running). Thus, one groups started with strength, the other with plyometrics, and last group with running tests, while after finishing the type of tests, the group continued in the following sequence: strength–plyometrics–running. Each group started at each test day with the same type of tests and in the same sequence to avoid a sequence effect. The specific warm-up for the strength, plyometric, and running tests was performing the exercise at sub-maximal intensities. After a group had tested their performance in one modality, they moved on to the next, until all tests were completed. The groups and order of tests were identical for the days of familiarization and pre- and post-test. All tests required three maximum effort attempts, with three minutes of rest between each. In the unilateral movements, performance was only tested for the right foot as the right foot performed the plant step in the COD tests which only included left turns [6]. The average performance of approved (not slipping or failing triggering timing gates) maximum effort attempts was included in the statistical analysis. All testing and training sessions were performed indoors in a controlled lab environment.

#### 2.2.1. Running Test

The athletes were instructed to complete each attempt of all the running tests as quickly as possible. All the running tests were performed on an indoor court surface (Taraflex Sport Evolution M 7.0 mm, Unisport, Vantaa, Finland) whereby time was measured with wireless timing gates (Brower Timing Systems, Salt Lake City, UT, USA). All running tests started from a standing start with the front foot placed 5 cm behind the first timing gate, preventing a false trigger. The height of the first pair of timing gates was 1 m, while the other pairs of timing gates were set at 1.2 m. After a signal from a researcher, the athlete performed the test on their own accord to prevent the introduction of a reactive component to the test. Time started when the athlete passed the first timing gate and stopped when they passed the last timing gate. Distance and velocity in the different running tests was measured and calculated with a wireless CMP3 distance sensor laser gun, which was placed 1.8 m behind the starting position of the test (Noptel Oy, Oulu, Finland), sampling at 2.56 KHz. The laser was adjusted to point at the athlete’s lower back while running. A contact mat (IR-Contactmat-ML6TJP02-870, Ergotest Innovation, Porsgrunn, Norway), which sends and reflects an infrared carpet and registers when the carpet is disrupted, was used to measure foot contact. Musclelab 10.5.69 (Ergotest innovation A. S, Porsgrunn, Norway) synchronized the laser gun and the contact mat to measure step kinematics such as average ground contact time, flight time, step length, step frequency and peak velocities, while horizontal braking force (N/Kg) and power (W/Kg) were measured in the braking test.

The COD tests for measuring force- and velocity-oriented COD performance consisted of a 10 m sprint before performing a turn oriented towards either force (180°) (Figure 1A) or velocity (45°) (Figure 1B) then re-accelerating 10 m to finish the test. Total (10 m + COD + 10 m) were defined as the performance variable, accompanied by partial times (first and last 10 m) to investigate which part of the test was improved. The pivot of the 45° COD was initiated after placing the plant step on a 0.8 m line, while in the 180° COD the line had to be crossed for an attempt to be approved. Intraclass correlation of the different COD performances revealed excellent reliability ranging from 0.921 to 0.933.

The sprint test was a 30 m straight-line sprint, whereby the laser gun measured peak velocities. The maximum horizontal braking test was adopted from Harper, et al. [31], with the use of an acceleration–deceleration test which is included in the Musclelab v10.5.69 software (Ergotest Innovation A.S, Porsgrunn, Norway). Athletes had to sprint 20 m as fast as possible and then initiate maximum deceleration after passing the line at the 20 m mark (Figure 1C) in which horizontal braking force (N/kg) and power (W) were measured. For an attempt to be approved, the velocity passing the 20 m mark had to be > 95% of their peak velocity in the 30 m straight-line sprint; this was automatically controlled for by the Musclelab software.

#### 2.2.2. Strength Performance

The strength tests included in the current study were a bilateral barbell back squat, a unilateral quarter squat performed in a smith machine, and a lateral barbell squat. The lateral and quarter squats required a knee-bend of 90° measured from the landmark’s trochanter major, patella, and the lateral malleolus (Figure 2A). The depth requirements for the bilateral back squat were to bend the knee until trochanter major was in line with the patella, making the femur 90° perpendicular to the surface (Figure 2B).

Performance in the strength tests was measured using relative strength (1-RM/body mass), which was estimated by the load–velocity relationship in Jovanović and Flanagan [32], using the best fit line of regression with three different data points for each individual athlete. The data points correspond to loads at different velocities (≈1, 0.8 and 0.5 m/s). Loads were matched from pre- to post-test, as a faster velocity with the same load corresponds to gains in strength. Athletes performed three repetitions at each load, whereby the average of the second and third repetition was used for statistical analysis, as the first repetition at lighter loads is often slower [33]. Thus, athletes were instructed to lift upwards as fast as possible. Velocity in the concentric phase of the lift was measured with a linear encoder sampling at 500 Hz (ET-Enc-02, Ergotest Technology AS, Porsgrunn, Norway).

#### 2.2.3. Plyometric Performance

The plyometric tests assessed were a unilateral countermovement jump for maximal height (Figure 3A), a unilateral skate-jump performed (Figure 3C) laterally for maximal length (cm), and a drop-jump from a drop height of 0.2 m (Figure 3B), whereby the reactive strength index (flight time/contact time) was the performance variable. The drop-jump and unilateral countermovement jump were performed with the athlete’s hands placed on their hip to prevent their arm swing from contributing to the upward motion. The athlete was required to land on their right foot in the unilateral countermovement jump.

When performing the drop-jump, athletes were instructed to “walk out” from the drop height to prevent increased drop height. Ground contact time and jump height were ascertained using a dual force plate (Ergotest Technology AS, Porsgrunn, Norway) with a sampling rate of 1000 Hz. The force plate registers contact time and flight time and calculates jump height with the use of flight time with the following equation: jump height = ½ × 9.81 × ${\left(\frac{\mathrm{flight}\;\mathrm{time}}{2}\right)}^{2}$. Athletes were free to use their arm swing in the skate-jump but had to manage the landing without moving their left foot, which performed the landing. Jump distance in the skate-jump was manually measured with a measuring tape to the closest 0.01 m.

#### 2.2.4. Training

By utilizing the pair-matched method of average COD performance in the pre-test, the subjects were randomly divided into two groups, namely either strength or plyometrics, prior to the training intervention. The data from the pre-test was used to calculate impulse (∆*mv* = ∫*Fdt*) for one repetition of each exercise, a method used in earlier research to normalize the training impulse of different training programs [29,34]. The strength and plyometric exercises trained were the same exercises which were tested from pre- to post-test; these were matched in movement patterns and the number of exercises performed bilaterally, unilaterally, and laterally.

Two weeks after the pre-test, the training intervention was initiated. The training intervention consisted of 1–2 training sessions a week for a total of 8 weeks, with 12 sessions in total. During the intervention, athletes were informed to not perform extra strength or plyometric training on the lower limbs, which did not follow regular handball training. Each training session was conducted prior to their regular training. A minimum of 48 h of rest was required between each session. One researcher controlled all the plyometric training sessions, while another controlled all the strength training sessions to provide technical guidance when training and providing motivation to obtain maximal effort.

Strength exercises were performed with a controlled eccentric phase, while the concentric phase required maximum concentric velocity. Load for the strength training was individualized based on the pre-test data, whereby athletes maximized the loads set at every session without compromising technique. The strength training program was periodized following a double linear progression with increments in load or the total number of reps performed. Both training protocols followed a linear increase in training volume with a de-load of the lower training volume in the final week of the intervention (Table 2). The plyometric exercises emphasized maximal jump height, while minimizing ground contact times. Contact mats were regularly used during plyometric training to provide the athletes with performance feedback, which also functioned as a motivational tool. The only exceptions in execution from testing to training were that the skate-jump was performed repeatedly from side-to-side during the training intervention, and both the left and right legs were trained unilaterally for all the unilateral exercises.

### 2.3. Statistical Analysis

The descriptive statistics are presented as mean ± standard deviation. An independent samples t-test was used to compare the two groups at baseline. To assess the effect of plyometric and strength training a 2 (pre- and post-test: repeated measurements) × 2 (group: plyometric, strength) ANOVA was performed. In addition, a one-way ANOVA with repeated measures per group was conducted to identify changes for each group. The assumption of normality was controlled for utilizing the Shapiro–Wilk test. When the assumption of normality was violated, the Kruskal–Wallis H test was employed. Effect size is presented by eta squared (η^2^) whereby 0.01 < η^2^ < 0.06 was defined as a small effect, 0.06 < η^2^ < 0.14 was defined as a medium effect and η^2^ > 0.14 corresponded to a large effect [35]. The level of significance was set at *p* < 0.05. All the tests were carried out in SPSS v.27 (IBM Corp., Armonk, NY, USA).

## 3. Results

No significant differences were observed between the two groups in any of the variables at baseline (t ≤ −1.68; *p* ≥ 0.1). Similarly, no significant group effects between the two training modalities from pre- to post-test in any of the tests (F ≤ 2.70; *p* ≥ 0.12; η^2^ ≤ 0.14) were found, except for the reactive strength index favoring the plyometric training group and the bilateral back squat favoring the strength training group (F ≥ 5.03; *p* ≤ 0.04; η^2^ ≥ 0.41) (Table 3).

However, when analyzing per group, a significant decrease in time was only observed in the strength training group in total time to complete the 180° COD and the last 10 m of the 180° COD (F ≥ 5.51; *p* ≤ 0.04; η^2^ ≥ 0.36). No significant changes in COD performances were observed in the plyometric training group (F ≤ 2.95; *p* ≥ 0.13; η^2^ ≤ 0.3). Dependent on timing points (first/last 10 m or 20 m total), only three to six out of eleven athletes in the strength training group decreased COD times in the 45° COD vs. four out of ten in the plyometric training group. In the 180° COD, however, seven to nine out of eleven athletes in the strength training group decreased COD times vs. six to seven out of ten in the plyometric training group (Figure 4 and Figure 5). Furthermore, the strength training group significantly decreased their 20 m and 30 m sprint times (F ≥ 10.57; *p* ≥ 0.01; η^2^ ≥ 0.51), while the plyometric training group significantly decreased their 30 m sprint time (F = 11.77; *p* < 0.01; η^2^ = 0.57). No significant changes were observed in horizontal braking force or power for either of the two groups (F ≤ 1.56; *p* ≥ 0.27; η^2^ ≤ 0.24) (Table 4).

The strength training group significantly increased their peak velocity from pre- to post-test in the 30 m sprint and 180° COD (F ≥ 7.09; *p* ≤ 0.03; η^2^ ≥ 0.44), while the plyometric training group increased their peak velocity and decreased their average contact time in the 30 m sprint (F ≥ 6.6; *p* ≤ 0.03; η^2^ ≥ 0.45) (Table 5).

## 4. Discussion

The purpose of the current study was to investigate the effect of strength vs. plyometric training upon developing force- and velocity-oriented COD performance in young female handball players. The main findings were that the strength training group significantly decreased the times of their total 20 m and last 10 m in the force-oriented COD, whilst approaching the COD at a higher peak velocity, while the plyometric training group experienced no significant improvements in COD performance (Table 4). The force-oriented COD improvements in the strength training group could be the result of improved knee flexor and hip extensor strength, as indicated by strength increases in the bilateral- and quarter squat. Earlier research suggests athletes unable to back squat 1.5 × body mass to benefit from improving strength in this exercise, which is furthermore suggested to be more specific towards enhancing force-oriented COD performance [9,14,16]. The athletes of the current study were able to squat 1× body mass in external load at baseline, which increased to 1.2× body mass in the strength training group.

The different squat variations largely target the knee flexors and hip flexors, which have been suggested to be of major importance in the acceleration phase [36], and also deceleration phase [16]. Consequently, the increase in relative strength observed at post-test in the strength training group could have increased the athletes’ ability to accelerate their own body mass, which also explains the higher peak velocity approaching the pivot in the force-oriented COD. Higher peak velocity approaching the eccentric braking phase of the COD could potentially potentiate the concentric re-acceleration after the pivot due to stored elastic energy in the lengthened muscle [37]. As such, the strength training group was able to change their momentum to the opposite direction faster at post-test, despite approaching the COD at a higher peak velocity (Table 5). Such a change in momentum demands a high impulse to create a propulsive force to redirect the athlete into the opposite direction of travel [16]. As such, increasing the ability of athletes’ lower limb muscles to produce net force positively influences COD performance as change in momentum is dictated by time for applied forces [17]. A change in momentum was observed by the reduced time to pivot, which could be a result of improved knee extensor strength promoting greater neuromuscular control in the contact phase of the COD [38], allowing a faster transition from the weight-acceptance phase to the concentric re-acceleration.

Accordingly, there could exist a transfer between the strength training program and the force-oriented COD manoeuvre as earlier research has found both bilateral and quarter squat performance to be associated with force-oriented COD performance in a similar population [23]. It is not possible to evaluate the isolated effect of the different exercises, but it could be speculated that the bilateral back squat shares similarities with the pivot in the force-oriented COD. This is because the pivot in a 180° COD is performed bilaterally with great knee and hip flexion [6], which is closest to the bilateral back squat in terms of the movement patterns of the exercises included in the current study. Similarly to the bilateral back squat, the pivot in the force-oriented COD is based on contact time [6,16], suggesting the need for a relatively long time to produce high amounts of forces to re-accelerate one’s own body mass, as the horizontal velocity of the body is zero in the weight-acceptance phase. Furthermore, a group effect was observed in relative strength in the bilateral back squat (Table 3).

No significant change in performance in the force-oriented COD was observed in the plyometric training group. The effect of the plyometric training program upon force-oriented COD performance in the studied population seems trivial based upon the individual responses (Figure 4 and Figure 5). The effect of strength training upon COD performance was expected based upon earlier research in female athletes. However, the lack of COD improvements in the plyometric training group is contradictive to earlier research in female athletes [9]. The lack of improvements in the force-oriented CODs in the plyometric training group could be because the plyometric exercises were not a sufficient stimulus to express the forces required to pivot in the force-oriented COD. Another explanation is that the total number of foot contacts in the training intervention were not a sufficient training stimulus to elicit targeted adaptions, as the number of foot contacts in the first sessions was relatively low (<50) [39]. Another explanation is that the number of foot contacts were not a sufficient training stimulus to elicit targeted adaptions. However, incorporating high amounts of foot contacts without an adaptive phase may increase the risk of injury [39]. Although research has evidenced that plyometric training is effective in developing force- and velocity-oriented COD performance in strength adapted men [29], athletes displaying lower levels of strength might benefit more from gaining strength prior to plyometric training [14] to prevent the plyometric training from producing slower returns. The assumption is in accordance with earlier research, whereby the plyometric exercises of the current study have been found to be associated with COD performance in strength adapted men [22], while the strength exercises were associated with COD performance in young female athletes expressing lower levels of relative strength [23]. However, the lack of improvements in the plyometric training group could also be explained by biological age, as the females in this study were young (17.1 years) in comparison to the aforementioned training intervention conducted in men (22.6 years). This is because complex exercises, such as the plyometric exercises included in this study might be less efficient in younger athletes due to less strength and motor control [9].

In the velocity-oriented COD, no significant changes were observed for either of the two groups. A possible explanation is that the exercises included in the two training programs mainly target the knee extensors and gluteus muscles. The exercises provide limited stimulus to the hamstring muscles and possibly the adductors, which have been found to be very active in the force-oriented COD [40], but even more so in velocity-oriented CODs [22]. Earlier research highlighted that the hamstring muscles play a major role in the deceleration phase of the COD [40,41,42,43,44]. Moreover, the force- and velocity-oriented CODs have been suggested to be separated by EMG activity in the adductor longus [6], displaying higher amplitude in the velocity-oriented CODs due to the adductors’ importance in hip stabilization [44,45].

Both groups significantly increased performance in different strength and plyometric tests (Table 3), indicating the existence of a dependency on similar abilities for performance enhancement. The finding was also observed in the 30 m sprint as both training groups significantly decreased their time to sprint 30 m and a large effect was observed for both groups across all timing points (5 m, 10 m, 20 m, and 30 m). The finding is in line with earlier research suggesting a transfer from quarter-squat training to sprint and countermovement jump performance [46], which improved in both groups. When evaluating the 30 m sprint test, it is likely that both groups improved their acceleration and straight line sprinting abilities, which are physical qualities indicated to be related to COD performance [5,40]. However, the neglection of training exercises targeting the hamstring muscles seems to have limited the deceleration abilities a priori the COD step. Although the plyometric training group seemingly improved performance in the horizontal braking test, the improved performance did not transfer to COD performance, possibly due to a different deceleration strategy in terms of the movement patterns and strength requirements of the adductor longus.

The study is limited by no control group and its number of subjects due to the logistical demands of completing a training intervention. Furthermore, due to the equipment available it was not possible to measure step kinematics in the velocity-oriented COD. However, no significant changes were observed in velocity-oriented COD performance, suggesting the step kinematics to be similar to baseline. Replication with greater participation and measurements of step kinematics in the velocity-oriented COD is warranted. Lastly, strength performance was for practical reasons measured utilizing the force-velocity relationship, which may differ from a true 1-RM. However, the same loads were used for both pre- and post-test, whereby higher velocities at the same loads corresponds to strength increases with great surety.

## 5. Conclusions

Only the strength training program was effective in developing force-oriented COD performance, possibly due to similarities in movement patterns of the pivot along with longer times to generate high forces. Athletes displaying lower levels of relative strength might benefit more from training strength prior to plyometric training to develop force-oriented COD performance. However, both the strength and plyometric training was indicated to be effective in enhancing performance in different strength, plyometric, and power tests in young female handball players.

Strength training of different squat variations targeting the knee flexors and hip extensors may be useful to improve force-oriented COD performance in young female handball players, before more specific training is required. However, the supplementation of training targeting the hamstrings to improve deceleration abilities should also be considered, which is possibly neglected with the usage of the exercises in the current study.

## Figures and Tables

**Figure 1 ijerph-19-06946-f001:**
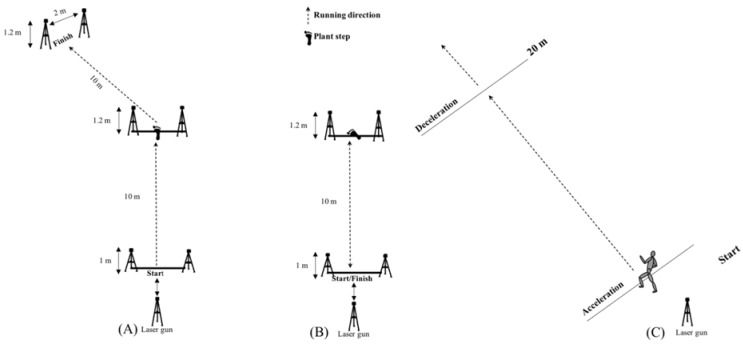
Set-up for maximum horizontal braking test (**A**), the 45° (**B**) and 180° (**C**) COD test.

**Figure 2 ijerph-19-06946-f002:**
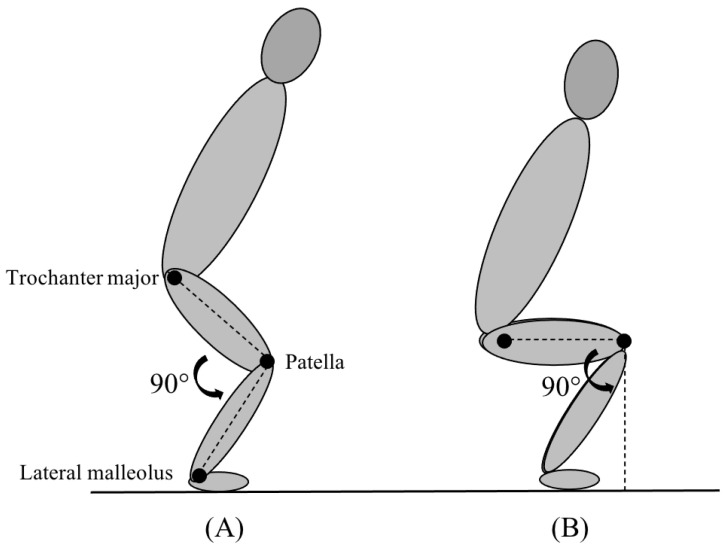
Depth requirements for the quarter squat and lateral squat (**A**) and the bilateral back squat (**B**).

**Figure 3 ijerph-19-06946-f003:**
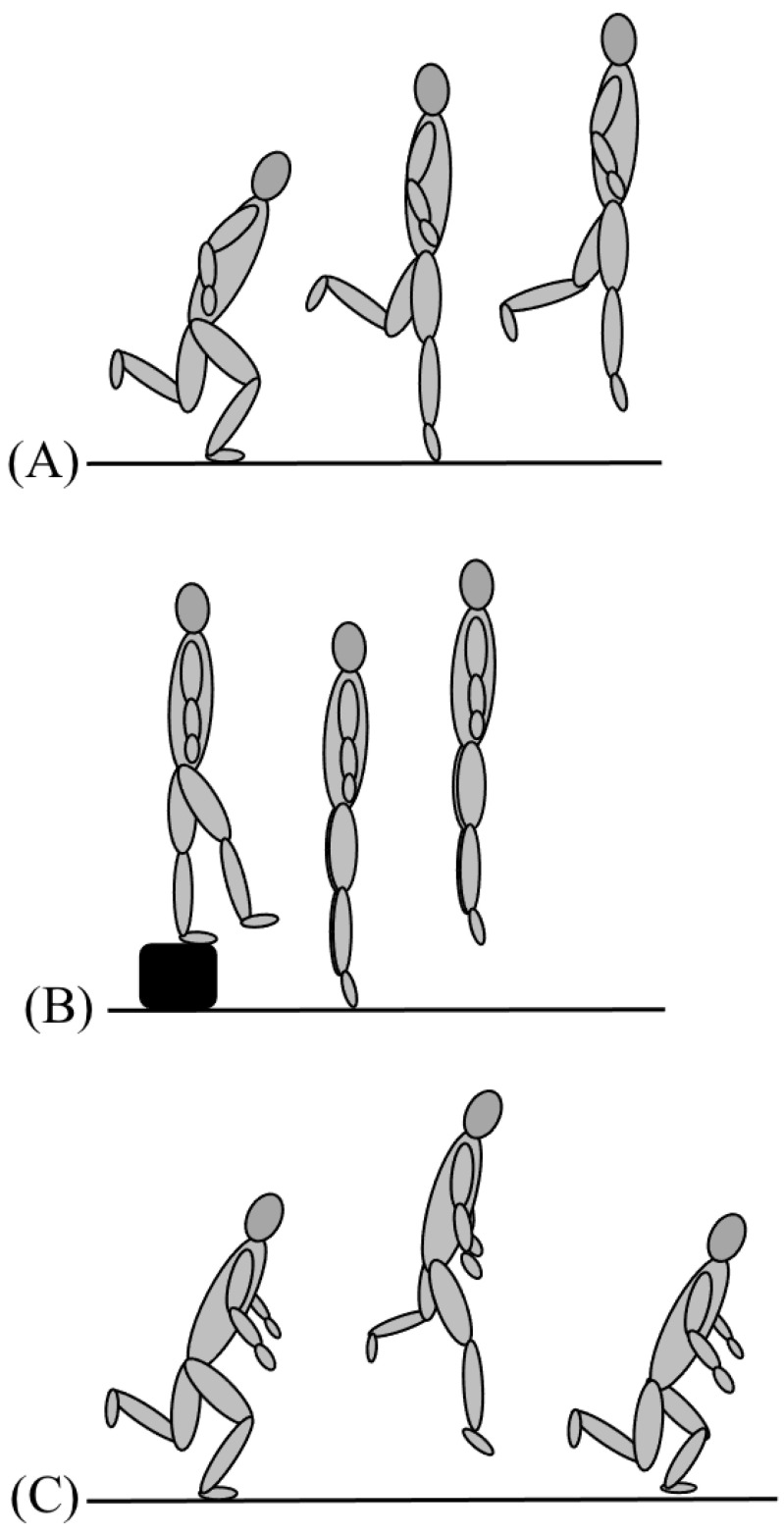
Illustration of the different phases of the different plyometric exercises. A unilateral countermovement jump: (**A**) countermovement, take-off, mid-air. (**B**) Drop-jump; “walk-out”, take-off, mid-air. (**C**) Skate-jump; countermovement, take-off in the lateral direction, impact with the alternating leg.

**Figure 4 ijerph-19-06946-f004:**
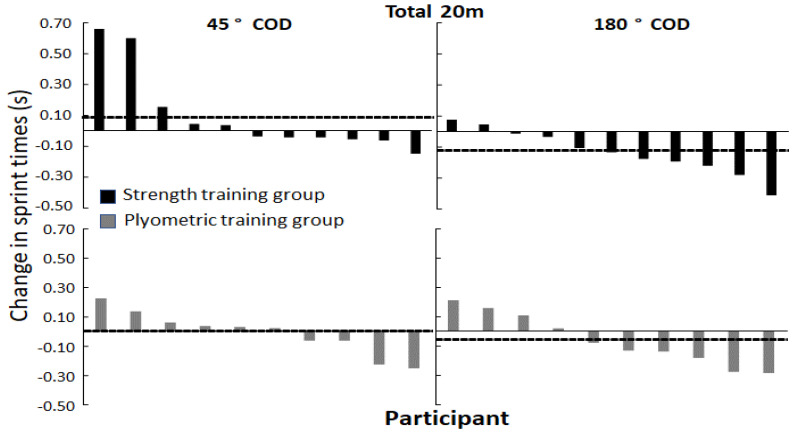
Individual responses displayed in seconds for both the strength and plyometric training groups in the 45° and 180° COD test at total COD test (20 m). Average change in performance within the group is marked by a dotted line.

**Figure 5 ijerph-19-06946-f005:**
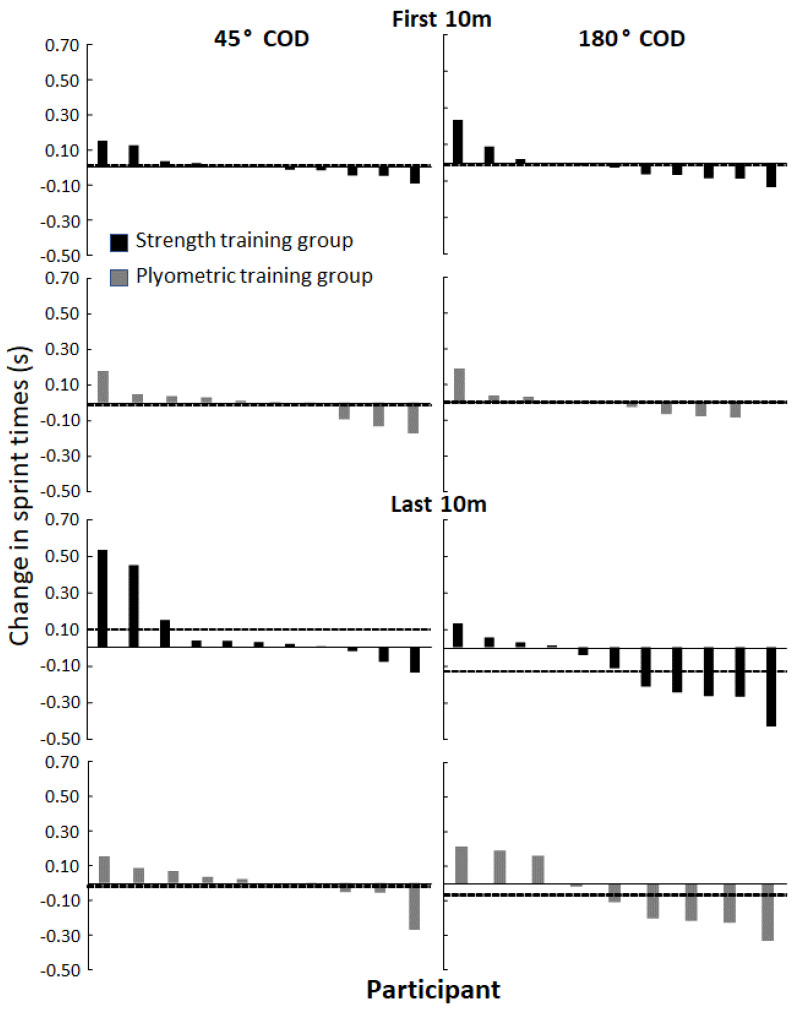
Individual responses displayed in seconds for both the strength and plyometric training groups in the 45° and 180° COD test at first and last 10 m. Average change in performance within the group is marked by a dotted line.

**Table 1 ijerph-19-06946-t001:** Descriptive information of the strength and plyometric training group.

	Age (Years)	Height (cm)	Body Mass (Kg)
**Strength training (n = 11)**	17.5 ± 2.3	169.2 ± 5.4	65.8 ± 5.9
**Plyometric training (n = 10)**	17.1 ± 2.4	173.1 ± 6.6	67.1 ± 9.3

**Table 2 ijerph-19-06946-t002:** Training program with the training impulse for the strength and plyometric group.

Training Day	Strength Training Group	Plyometric Training Group	Repetitions/Ground Contacts × Sets	Training Impulse (Ns) per Session	Rest between Sets
**1 and 2**	Bilateral Squat	Drop jump	4 × 2/1 × 6	≈4550	3–5 min/>2 min
	Unilateral Squat	Unilateral CMJ	4 × 2/1 × 5		3–5 min/>2 min
	Lateral Squat	Skate-jump	4 × 2/6 × 2		3–5 min/>2 min
**3 and 4**	Bilateral Squat	Drop jump	5 × 2/1 × 8	≈5700	3–5 min/>2 min
	Unilateral Squat	Unilateral CMJ	5 × 2/1 × 5		3–5 min/>2 min
	Lateral Squat	Skate-jump	5 × 2/6 × 3		3–5 min/>2 min
**5 and 6**	Bilateral Squat	Drop jump	6 × 2/1 × 8	≈6880	3–5 min/>2 min
	Unilateral Squat	Unilateral CMJ	6 × 2/1 × 8		3–5 min/>2 min
	Lateral Squat	Skate-jump	6 × 2/6 × 3		3–5 min/>2 min
**7 and 8**	Bilateral Squat	Drop jump	5 × 3/1 × 10	≈8500	3–5 min/>2 min
	Unilateral Squat	Unilateral CMJ	5 × 3/1 × 9		3–5 min/>2 min
	Lateral Squat	Skate-jump	5 × 3/6 × 4		3–5 min/>2 min
**9 and 10**	Bilateral Squat	Drop jump	6 × 3/1 × 12	≈10300	3–5 min/>2 min
	Unilateral Squat	Unilateral CMJ	6 × 3/1 × 11		3–5 min/>2 min
	Lateral Squat	Skate-jump	6 × 3/6 × 5		3–5 min/>2 min
**11**	Bilateral Squat	Drop jump	4 × 3/1 × 8	≈6880	3–5 min/>2 min
	Unilateral Squat	Unilateral CMJ	4 × 3/1 × 8		3–5 min/>2 min
	Lateral Squat	Skate-jump	4 × 3/6 × 3		3–5 min/>2 min
**12**	Bilateral Squat	Drop jump	3 × 2/1 × 5	≈3430	3–5 min/>2 min
	Unilateral Squat	Unilateral CMJ	3 × 2/1 × 4		3–5 min/>2 min
	Lateral Squat	Skate-jump	3 × 2/1 × 4		3–5 min/>2 min

CMJ = Countermovement jump; repetitions for the unilateral and lateral squat and the unilateral CMJ is presented per leg, whereby rest time was 1 min between each leg.

**Table 3 ijerph-19-06946-t003:** Descriptive statistics of performance changes from pre- to post-test for both groups in the different strength and plyometric tests.

	Strength Training Group				Plyometric Training Group			
	Pre	Post	Change (%)	Effect Size (η^2^)	Pre	Post	Change (%)	Effect Size (η^2^)
**Strength tests**								
Bilateral squat (Kg/BM)	0.98 ± 0.29	1.21 ± 0.34	20.07 *	0.52 ‡	0.98 ± 0.32	0.99 ± 0.26	2.35	<0.01
Quarter squat (Kg/BM)	0.58 ± 0.37	0.83 ± 0.31	40.39	0.57	0.43 ± 0.11	0.64 ± 0.14	49.88 *	0.57
Lateral squat (Kg/BM)	0.51 ± 0.11	0.59 ± 0.23	12.56	0.13	0.59 ± 0.23	0.51 ± 0.14	10.12	0.11
**Plyometric tests**								
Drop jump (RSI)	0.82 ± 0.22	0.92 ± 0.24	11.87 *	0.48 ‡	0.80 ±0.25	1.27 ± 0.30	57.9 *	0.68
Countermovement jump (cm)	9.3 ± 2.2	10.4 ± 2	11.28 *	0.46	10.0 ± 2.3	12 ± 3.1	20.48 *	0.60
Skate jump (cm)	159.1 ± 15.3	164 ± 22.6	3.07	0.08	157 ± 10.4	171.7 ± 5.7	9.36 *	0.67

* Indicates a significant difference from pre- to post-test measurements at a *p* < 0.05 level. ‡ Indicates a group effect at a *p* < 0.05 level. BM = Body mass. RSI = Reactive strength index.

**Table 4 ijerph-19-06946-t004:** Descriptive statistics of performance changes in the change of direction test, 30 m sprint and horizontal braking test for both the strength and plyometric training group.

Strength Training Group				
	Pre (Mean ± STD)	Post (Mean ± STD)	Change (%)	Effect Size (η^2^)
First 10 m 45° (s)	2.22 ± 0.09	2.23 ± 0.1	0.45	0.02
20 m total 45° (s)	3.95± 0.02	4.04 ± 0.36	2.23	0.13
Last 10 m 45° (s)	1.73 ± 0.11	1.82 ± 0.24	4.95	0.18
First 10 m 180° (s)	2.42 ± 0.14	2.41 ± 0.14	−0.41	0.02
20 m total 180° (s)	5.30 ± 0.27	5.16 ± 0.27	−2.71 *	0.49
Last 10 m 180° (s)	2.90 ± 0.24	2.77 ± 0.23	−4.69 *	0.36
5 m sprint (s)	1.36 ± 0.08	1.33 ± 0.09	−2.21	0.32
10 m sprint (s)	2.24 ± 0.11	2.21 ± 0.13	−1.34	0.23
20 m sprint (s)	3.80 ± 0.2	3.72 ± 0.2	−2.11 *	0.51
30 m sprint (s)	5.31 ± 0.33	5.19 ± 0.27	−2.26 *	0.56
Braking power (W/Kg)	−10.29 ± 1.98	−10.25 ± 2.03	0.39	<0.01
Braking Force (N/Kg)	−2.9 ± 0.76	−2.74 ± 0.86	−5.52	0.01
**Plyometric Training Group**				
	**Pre (Mean ± STD)**	**Post (Mean ± STD)**	**Change (%)**	**Effect Size (η^2^)**
First 10 m 45° (s)	2.24 ± 0.10	2.23 ± 0.07	−0.45	0.02
20 m total 45° (s)	3.99 ± 0.19	3.97 ± 0.15	−0.5	0.01
Last 10 m 45° (s)	1.75 ± 0.12	1.75 ± 0.11	0	<0.01
First 10 m 180° (s)	2.41 ± 0.09	2.41 ± 0.12	0	0.01
20 m total 180° (s)	5.31 ± 0.25	5.24 ± 0.23	−1.34	0.06
Last 10 m 180° (s)	2.91 ± 0.27	2.85 ±0.17	−2.11	0.30
5 m sprint (s)	1.37 ± 0.06	1.35 ± 0.08	−1.46	0.19
10 m sprint (s)	2.27 ± 0.11	2.21 ± 0.10	−2.64	0.30
20 m sprint (s)	3.79 ± 0.15	3.74 ± 0.17	−1.32	0.29
30 m sprint (s)	5.32 ± 0.22	5.21 ± 0.23	−2.07 *	0.57
Braking power (W/Kg)	−9.65 ± 2.46	−10.49 ± 1.53	8.7	0.24
Braking Force (N/Kg)	−3.01 ± 0.65	−3.23 ± 0.42	7.3	0.22

* Indicates a significant difference from pre- to post-test measurements at a *p* < 0.05 level.

**Table 5 ijerph-19-06946-t005:** Descriptive statistics of changes in step kinematics from pre- to post-test measurements in both groups.

	Strength Training Group				Plyometric Training Group			
	Pre	Post	Change (%)	Effect Size (η^2^)	Pre	Post	Change (%)	Effect Size (η^2^)
**Peak velocities (m/s)**								
30 m print	6.79 ± 0.5	6.9 ± 0.4	1.62 *	0.44	6.75 ± 0.32	6.89 ± 31.7	2.07 *	0.45
180°	5.26 ± 0.3	5.4 ± 0.32	2.66 *	0.70	5.23 ± 0.17	5.29 ± 0.27	1.15	0.11
**Contact time (ms)**								
30 m sprint	169.7 ± 17.8	167.5 ± 19.4	−0.89	0.07	175.6 ± 10.7	168.3 ± 13.2	−4.16 *	0.49
180°	197.5 ± 22.2	195.8 ± 21	−0.86	0.02	201.3 ± 14.14	201.25 ± 11	−0.02	<0.01
**Flight time (ms)**								
30 m sprint	100.7 ± 13.2	99.6 ± 12.6	−1.09	0.01	102 ± 9.18	101.9 ± 107.9	−0.15	<0.01
180°	67.8 ± 15.1	66.2 ± 15	−2.36	0.03	65.42 ± 70.63	70.63 ± 14.69	7.96	0.15
**Step length (m)**								
30 m sprint	1.56 ± 0.11	1.58 ± 0.13	1.28	0.02	1.65 ± 0.11	1.59 ± 0.08	−3.64	0.27
180°	1.22 ± 0.11	1.22 ± 0.05	0	0.01	1.21 ± 0.06	1.23 ± 0.09	1.79	0.04
**Step frequency (n/time**)								
30 m sprint	3.78 ± 0.31	3.81 ± 0.33	0.79	0.05	3.67 ± 0.11	3.78 ± 0.24	3	0.28
180°	3.79 ± 0.32	3.87 ± 0.3	2.11	0.16	3.75 ± 0.17	3.73 ± 0.23	−0.53	0.03
**Time to pivot (ms)**								
180°	1116.5 ± 191.9	1005.4 ± 100.4	−9.95	0.23	1131.5 ± 187.5	9995.2 ± 142.3	−12.05	0.10

* Indicates a significant difference from pre- to post-test measurements at a *p* < 0.05 level.

## Data Availability

The raw data supporting the conclusions of this article will be made available by the authors without undue reservation.

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
