# Peer review of "Effect of Strength vs. Plyometric Training upon Change of Direction Performance in Young Female Handball Players"

_ijerph, 2022, doi:10.3390/ijerph19116946_

Round 1

Reviewer 1 Report

This study examines the effect of strength vs. plyometric training on change of direction performance in young female handball players. The study is very intriguing and relatively comfortable to follow and understand. Methods are appropriate and thorough. Statistical analysis was conducted very precisely, with results well presented. Furthermore, Introduction and Discussion were written competently. However, there are some comments and raised issues below, which I would like to get an answer from the authors. They are mainly related to Methods and Results.

Methods:

One of the main flaws of this study is the absence of the control group. Authors should thoroughly elaborate on why this is the case in the Methods section and list this as a study limitation. 
The testing protocol is not obviously presented. I would suggest that authors create maybe a chart or separate section (i.e., experiment protocol) to explain this. 

Why was the drop jump performed from the 20cm box? This height seems relatively low?

Can the authors elaborate on why they were waiting two weeks after the pretest to start with the training? This period is usually one week or just several days long.

Results:

Figure 4 has poor visibility. Please either increase the graph resolution or split this graph in two.

Please indicate on the graph what is the measurement on the x-axis.

Overall I am not entirely pleased with the visual presentation of the results. There are more tables than graphs, which is difficult to follow (sometimes even dull) for readers. My recommendation is to present the most critical findings in figures (where possible).

Author Response

 We want to thank the reviewer for reviewing the manuscript. We hae changed the manuscript according to the comments of the reviewer and think that it is now suitable for publication.

Reviewer 1

This study examines the effect of strength vs. plyometric training on change of direction performance in young female handball players. The study is very intriguing and relatively comfortable to follow and understand. Methods are appropriate and thorough. Statistical analysis was conducted very precisely, with results well presented. Furthermore, Introduction and Discussion were written competently. However, there are some comments and raised issues below, which I would like to get an answer from the authors. They are mainly related to Methods and Results.

Methods:

One of the main flaws of this study is the absence of the control group. Authors should thoroughly elaborate on why this is the case in the Methods section and list this as a study limitation. 

Lack of control group has now been added as a limitation. The authors agree that a control group would be of great value. However, the inclusion of a control group would reduce the participants in each group (n=7), significantly reducing statistical power to investigate the main research question (strength vs plyometric training). No control group is often a limitation in training interventions, and number of participants is a common reason why. Furthermore, the research question is also just a comparison of strength vs. plyometrics training and thereby a control group is not necessary to compare with. The authors are under the impression that this elaboration is not crucial due to its usuality, although, it can be included in the methods section if the reviewer think it is needed.

The testing protocol is not obviously presented. I would suggest that authors create maybe a chart or separate section (i.e., experiment protocol) to explain this. 

We have explained the testing protocol (especially the testing sequence) a bit more in the methods now. The athletes were randomized into three groups at familiarisation day. Subsequently, at familiarisation day, the three groups were randomly assigned to start to test performance in one of the type of tests (strength, plyometrics or running). Thus, one groups started with strength, the other with plyometrics and last group with running tests, while after finishing the type of tests, the group continued in the following sequence: strength – plyometrics – running. Each group started at each test day with the same type of tests and in the same sequence to avoid a sequence effect. We hope it is now easy to follow for the reader.

Why was the drop jump performed from the 20cm box? This height seems relatively low?
The height set was based on an earlier study conducted in our lab in a similar population (Falch et al, 2021). When conducting a pilot, several subjects reported discomfort from higher drop heights. When conducting the aforementioned study, the heights of >0.2m were observed to be to challenging as the subject’s strength limited important aspects from greater drop heights, such as: short foot contacts for a fast SSC (<0.25s), preventing the heel from touching the ground, preventing great knee- flexion and valgus etc. These observations led us to conclude that the height was reasonable as there was great room for improvement, while reducing the risk of injury by pushing the drop height in a training intervention. This is because plyometric training exceeding the participants ability, with progressions such as drop height (which was standardised to normalise training impulse), could increase the risk of injury (Bedoya et al, 2015). The height set was also set to avoid reducing the RSI-value for many of the participants, indicating less optimal drop height for this cohort on average (Lloyd et al, 2011). Based upon the results of the current study, the drop height was seemingly sufficient to enhance RSI.

Falch, H. N., Kristiansen, E. L., Haugen, M. E., & van den Tillaar, R. (2021). Association of Performance in Strength and Plyometric Tests with Change of Direction Performance in Young Female Team-Sport Athletes. Journal of Functional Morphology and Kinesiology6(4), 83.

Bedoya, A. A., Miltenberger, M. R., & Lopez, R. M. (2015). Plyometric training effects on athletic performance in youth soccer athletes: a systematic review. The Journal of Strength & Conditioning Research29(8), 2351-2360.

Lloyd, R. S., Meyers, R. W., & Oliver, J. L. (2011). The natural development and trainability of plyometric ability during childhood. Strength & Conditioning Journal33(2), 23-32.

Can the authors elaborate on why they were waiting two weeks after the pretest to start with the training? This period is usually one week or just several days long.
The authors agree with the reviewers’ point of view. The training intervention would be initiated earlier after the pre-test, if it was possible. The two weeks before initiating the training intervention was due to logistical challenges, as the training intervention (and testing) had to be convenient for the team participating in this study and their overall training schedule.

Results:

Figure 4 has poor visibility. Please either increase the graph resolution or split this graph in two.
We fully agree with the reviewer that this figure is of poor visibility. That is why we, after recommendation of the reviewer split it in two figures, which are not blurry.

Please indicate on the graph what is the measurement on the x-axis.

In the x-axis it is participant 1-10 (11) as mentioned in the legend individual changes from pre-to post. We hope it is clear now with the two figures.

Overall I am not entirely pleased with the visual presentation of the results. There are more tables than graphs, which is difficult to follow (sometimes even dull) for readers. My recommendation is to present the most critical findings in figures (where possible).

We have chosen to use mostly tables since we have very much information that perhaps could explain why COD enhanced in strength training group and not so much in plyometric group. We also like figures more than tables. However, due to the number of tests we did and the information we had from the test we found it better to stick to tables instead of showing more figures. We already have 5 tables and 5 figures, which we think is more than enough. The most important information we have shown in figure 4 and 5 in which the individual data was shown of the change in COD. We hope that the reviewer understands our point of view.

Reviewer 2 Report

The purpose of this paper was to investigate and compare the effects of strength training and plyometric training on change of direction in young female handball players. The strengths of this paper include clear writing style and use of tables/ charts. The main weakness of this paper was that the nonsignificant findings, especially with the plyometric training, made it difficult to apply the results to performance training. There were some good implications about the benefits of strength training in the discussion section though. The citations were current and appropriate. There were several limitations to the study (equipment, number of subjects), but all were discussed in the paper.  

Introduction:

-Around line 74- add a sentence or two about why you chose young female athletes specifically

-Line 93- you discussed the lack of research with team sports- but why did you choose to use handball? 

Methods

-Diagrams very helpful

Discussion

- line 368- discuss what is the ideal # of foot contacts in training

- line 371- is there any evidence that you can cite of plyometric training increasing injury in this population?

- lines 381-384- consider going into more detail about when plyometric training would start having benefits

Overall consider more discussion on why you tested females and how these results will benefit female athletes specifically. 

Author Response

We want to thank the reviewer for reviewing the manuscript. We have changed the manuscript according to the comments of the reviewer and think that it is now suitable for publication.

Reviewer 2

The purpose of this paper was to investigate and compare the effects of strength training and plyometric training on change of direction in young female handball players. The strengths of this paper include clear writing style and use of tables/ charts. The main weakness of this paper was that the nonsignificant findings, especially with the plyometric training, made it difficult to apply the results to performance training. There were some good implications about the benefits of strength training in the discussion section though. The citations were current and appropriate. There were several limitations to the study (equipment, number of subjects), but all were discussed in the paper.  

Introduction:

-Around line 74- add a sentence or two about why you chose young female athletes specifically
The following sentence has been added:
“The current knowledge of the effect of strength- and plyometric training upon enhancing COD performance is limited and greatly stemmed from training interventions conducted in male soccer players (Falch et al, 2019).”
The authors believe the sentence, in conjunction with the following lines provides rationale for conducting the study in a young population displaying lower levels of relative strength, but the rationale can be further elaborated if the reviewer think it is needed.

-Line 93- you discussed the lack of research with team sports- but why did you choose to use handball? 

We did choose handball, just by the reason that we had access to so many players in this sport and not in another team sport as like soccer at this level. Furthermore, we have done several studies in handball and thereby it is easier to compare interventions when using same type of group / sport.

Methods

-Diagrams very helpful

Thank you

Discussion

- line 368- discuss what is the ideal # of foot contacts in training
Line 368 is now rephrased with a citation to constitute that the volume of the first sessions was relatively low (<50 ground contacts) as the ideal number of foot contacts will be context dependent. A citation has also been added to support our evaluation of progressively increasing the number of foot contacts.

- line 371- is there any evidence that you can cite of plyometric training increasing injury in this population?
Plyometric training in isolation does not substantially increase the risk of injury in comparison to other training modalities. Introducing a new training stimulus (such as plyometric training) with a high training frequency in combination with a high volume/intensity on top of normal training load could substantially increase the risk of injury. Since plyometric training was introduced on top of normal training and furthermore performed at maximum levels of effort, training volume was progressively increased to prevent injuries.

- lines 381-384- consider going into more detail about when plyometric training would start having benefits
We thank the reviewer for a good perspective, leading to an important point. Plyometric training as a training modality is indicated in the literature to be useful across different age groups. However, appropriate exercises might vary with regards to complexity and baseline abilities for the specific population. The sentence has therefore been rephrased to:
“This is because complex exercises, such as the plyometric exercises included in this study might be less efficient in younger athletes due to less strength and motor control [1]”.

Overall consider more discussion on why you tested females and how these results will benefit female athletes specifically.
The authors agree that sex specific adaptations are a very interesting topic. Unfortunately, we believe our data is more suited for providing “proof-of-concept”, with valuable information regarding training considerations of athletes age and baseline levels of relative strength. However, we do not believe it is suitable for discussing sex specific benefits based on our measurements, since we only measured women. The population selected was young female athletes to investigate if athletes displaying lower levels of relative strength (or potentially motor control as a result of biological age) would display different results when compared to a similar study conducted in older, strength adapted men (Rædergård et al, 2020). As such, we believe sex related extrapolations must be done with caution.

Rædergård, H. G., Falch, H. N., & Tillaar, R. V. D. (2020). Effects of strength vs. plyometric training on change of direction performance in experienced soccer players. Sports8(11), 144.

  1. Falch, H. N.; Rædergård, H. G.; van den Tillaar, R., Effect of different physical training forms on change of direction ability: a systematic review and meta-analysis. Sports Med-open 2019, 5, (1), 1-37.

Round 2

Reviewer 1 Report

I don't have any further requests. Thank you for answering all of my questions and acknowledging all raised issues.